# Improving Mechanical Properties for Extrusion-Based Additive Manufacturing of Poly(Lactic Acid) by Annealing and Blending with Poly(3-Hydroxybutyrate)

**DOI:** 10.3390/polym11091529

**Published:** 2019-09-19

**Authors:** Sisi Wang, Lode Daelemans, Rudinei Fiorio, Maling Gou, Dagmar R. D’hooge, Karen De Clerck, Ludwig Cardon

**Affiliations:** 1Centre for Polymer and Material Technologies (CPMT), Department of Materials, Textiles and Chemical Engineering, Ghent University, 9052 Zwijnaarde, Belgium; Sisi.Wang@ugent.be (S.W.); Rudinei.Fiorio@UGent.be (R.F.); 2Centre for Textiles Science and Engineering (CTSE), Department of Materials, Textiles and Chemical Engineering, Ghent University, 9052 Zwijnaarde, Belgium; Lode.Daelemans@UGent.be (L.D.); Dagmar.Dhooge@UGent.be (D.R.D.); 3State Key Laboratory of Biotherapy and Cancer Center, West China Hospital, Sichuan University, Chengdu 610041, China; goumaling@scu.edu.cn; 4Laboratory for Chemical Technology (LCT), Department of Materials, Textiles and Chemical Engineering, Ghent University, 9052 Zwijnaarde, Belgium

**Keywords:** extrusion-based additive manufacturing, poly(lactic acid), poly(3-hydroxybutyrate), annealing, notched impact strength

## Abstract

Based on differential scanning calorimetry (DSC), X-ray diffraction (XRD) analysis, polarizing microscope (POM), and scanning electron microscopy (SEM) analysis, strategies to close the gap on applying conventional processing optimizations for the field of 3D printing and to specifically increase the mechanical performance of extrusion-based additive manufacturing of poly(lactic acid) (PLA) filaments by annealing and/or blending with poly(3-hydroxybutyrate) (PHB) were reported. For filament printing at 210 °C, the PLA crystallinity increased significantly upon annealing. Specifically, for 2 h of annealing at 100 °C, the fracture surface became sufficiently coarse such that the PLA notched impact strength increased significantly (15 kJ m^−2^). The Vicat softening temperature (VST) increased to 160 °C, starting from an annealing time of 0.5 h. Similar increases in VST were obtained by blending with PHB (20 wt.%) at a lower printing temperature of 190 °C due to crystallization control. For the blend, the strain at break increased due to the presence of a second phase, with annealing only relevant for enhancing the modulus.

## 1. Introduction

Additive manufacturing (AM), or 3D printing, enables the production of physical objects through the decomposition of successive layers of material. AM using polymers has been exploited in multiple, innovative ways to produce materials and functional devices [1,2]. For extrusion-based AM (EAM), one major drawback is the lack of commercially available polymers that can be used in a similar way as in well-established processes, such as injection molding and conventional extrusion. The most frequently used materials for AM are thermoplastic polymers acrylonitrile butadiene styrene (ABS) and poly(lactic acid) (PLA) [2,3]. A considerable number of studies have also examined thermoplastic filaments based on polycarbonate [4], polystyrene [5], polycaprolactone [6], polybutylene terephthalate [7], and blends of different thermoplastics [8,9].

The performance of AM-fabricated thermoplastic parts is often unsatisfactory due to intrinsic limitations in the mechanical properties of the starting material. Several methods can be used to increase the performance: (i) reinforcing the starting materials with high strength fibers or fillers [7,10,11,12]; (ii) polymer annealing [13,14,15,16,17]; (iii) blending with a nucleating agent [18,19,20] or polymers, such as polyhydroxyalkanoates (PHAs) [8,21,22,23], which can act as crystallization controllers. Regardless of the approach used to increase the performance, consideration should also be given to controlling the layer thickness and infill density [24,25].

In general, annealing increases diffusion between distinct particle morphologies, bonding of beads, and crystallinity [10,16,25]. The latter plays a specific and important role in semi-crystalline polymers likely by improving the mechanical and thermal stability of the resulting product [13,17,26,27,28,29]. Notably, most studies illustrate the relevance of annealing for polymers produced by injection or compression molding [13,14,15,16,17]; however, there are relatively few reports concerning annealing for AM fabricated parts, highlighting that further research should be conducted in this area [10,24,25,29]. For example, Torres et al. [24] examined the influence of layer thickness, infill density, and post-treatment time at 100 °C on the torsion properties of EAM PLA components. They found that the layer thickness and infill density were highly important for strength optimization, with the use of heat treatment implementation slightly improving the properties of the resulting product. Wang et al. [25] also improved the impact strength of PLA using an EAM method involving a 0.2 mm layer thickness to produce fewer voids (relative to a 0.4 mm layer thickness) during the printing stage, and showed that a 160 °C bed temperature could mimic annealing to produce a higher degree of crystallinity.

As annealing requires post-treatment, the printed parts may deform slightly and, in turn, lose their dimensional accuracy. To address this issue, nucleating substances, such as bio-based and biodegradable PHAs have been applied to act as crystallization controllers and increase the mechanical thermal properties of the product [8,22,23]. In many studies, PLA/PHA blends are prepared by dissolving the components in a solvent to cast a film [30,31]. However, the application of this biodegradable blend as “ink” for 3D printing remains limited. Gonzalez Ausejo et al. [23] blended PLA with PHA containing 3-hydroxybutyrate (HB) and 3-hydroxyvalerate (HV) units. The resulting 3D printed material exhibited favorable mechanical properties (when printed in the Z direction), thermal stabilities, and cell viability, which highlights the potential usefulness of this blend for tissue engineering applications and EAM technology. In our recent communication, we reported that poly(3-hydroxybutyrate) (PHB), the most commonly available PHA on the market, enhanced PLA crystallinity during 3D printing, provided that the correct melt flow index (MFI) was used [8]. However, this study did not focus on the mechanical properties of printed PLA/PHB parts.

Here, we therefore evaluated how the mechanical and thermal properties of PLA printed parts could be improved using a wide range of experimental characterization techniques, including scanning calorimetry (DSC), X-ray diffraction (XRD) analysis, polarizing microscope (POM), and scanning electron microscopy (SEM), to examine the optimal processing conditions and the effects of variations in PHB blending and annealing, or a combination of both. Notably, PLA and PHB are bio-degradable; thus, the material can freely enter the eco-cycle. Hence, the present study closed the gap in applying conventional processing optimizations for the field of 3D printing, taking into account sustainability.

## 2. Materials and Methods

### 2.1. Materials and Processing

Poly(lactic acid) (PLA, Ingeo™ 3D850, NatureWorks, Minnetonka, MN, USA) pellets contained 0.5% D-isomer, and poly(3-hydroxybutyrate) (PHB) pellets were obtained from Biomer (P304) (Krailling, Germany). MFI, melt temperature, and thermogravimetric analysis data are shown in Appendix A. The materials were dried overnight at 50 °C before processing.

The printing of PLA and blends was targeted. A single screw extrusion apparatus was used for the production of filaments having a diameter of 1.75 ± 0.05 mm that is suitable for an EAM 3D printer [11,32]. For PLA, virgin PLA pellets were first added to a single screw extruder (Brabender PL2000, Cologne, Germany) with a screw diameter of 19 mm and an L/D of 25 to make the PLA filaments directly. The temperature varied from 180 to 210 °C with a screw speed set at 30 rpm and filament hauling speed at 8 m min^−1^. For blends, PLA/PHB blends (80/20 by weight) [33] were first converted into granules by twin-screw extrusion (Coperion ZSK18, Stuttgart, Germany; co-rotating screws with screw diameter of 18 mm and L/D of 40) with a barrel temperature that ranged from 155 °C to 200 °C and a screw speed of 120 rpm. The extrudates were water-cooled and cut into granules for the subsequent filament processing step in the single screw extruder (see Scheme 1a). The temperature profile was 170 to 200 °C for the PLA/PHB blend, and the screw and hauling speed were identified as those for the PLA case.

Dog bones (type 1BA, ISO527) and impact bars (100 × 10 × 4 mm, ISO 179) of virgin and blended PLA were generated using a Felix 3.0 printer with 3D printing parameters shown in Table 1. PLA was typically printed with a set-point of 210 °C, and the PLA/PHB blend was printed at 190 °C, a temperature at which PHB has a good flowability (MFI data in Appendix A), and the mechanical properties for the materials were optimal (Appendix A). Samples were annealed for different times (0.5 h, 1 h, and 2 h) at 80 °C and 100 °C. The treatment temperature was chosen according to the crystallization property of pure PLA and the PLA/PHB blend that have a cold crystallization peak (*T*_cc_) of ca. 100 °C and 80 °C, respectively (values were determined from DSC curves; see below). Samples were printed in good quality and showed no wrapping after annealing (Appendix A). The parts were then stored under standard environmental conditions (23 °C, 50% relative humidity) until characterization was conducted.

### 2.2. Characterization

Differential scanning calorimetry (DSC) measurements were conducted with a NETZSCH 204 F1 instrument to determine the effect of annealing on crystallinity. Samples were heated from room temperature to 200 °C at a heating rate of 10 °C min^−1^. The cold crystallization peak (*T*_cc_) and cold crystallization enthalpy (Δ*H*_cc_), as well as the melting peak (*T*_m_) and the melting enthalpy (Δ*H*_m_), were recorded. The crystallinity of the pure material was calculated by *X*_c_ = ((Δ*H*_cc_ + Δ*H*_m_)/Δ*H*_m_^0^) × 100%, with Δ*H*_m_^0^ the melting enthalpy of the 100% crystalline polymer (93 J g^−1^ [34] and 146 J g^−1^ [35] for PLA and PHB, respectively). DSC data were listed in Appendix A.

The crystalline structures of annealed and non-annealed samples were investigated using a Thermo Scientific ARL X’TRA X-ray diffractometer (XRD) (Portland OR, USA) with Cu-Kα radiation, an angle range (2θ) from 2° to 35°, and a scanning rate of 0.02° s^−1^.

Microscopic observation of the internal structure of samples was performed to visualize the quality of the material deposition, the presence of interior voids, and bonding between filaments and layers. To facilitate characterization with a polarizing microscope (POM) (Keyence VHX-6000, Mechelen, Belgium), a thin section (10 µm thick) from the middle of a printed impact bar sample mounted on a glass slide (Scheme 1b; xz orientation) was cut using a microtome (Leica RM2245). The cryo-fracture surface cross-sections of the impact bars were analyzed using a scanning electron microscope (SEM, PHENOM AO-IA-005, Eindhoven, Netherlands). Samples were immersed in liquid nitrogen at a temperature of –195 °C before breakage.

Tensile measurements (x-direction) were performed on Instron 5565 tensile machine according to the ISO 527 standard, using a 2620–603 Instron extensometer with a gauge length of 25 mm. A tensile rate of 1 mm min^−1^ was applied until 0.3% strain to determine Young’s modulus. Thereafter, a rate of 5 mm min^−1^ was used until the material broke. The average of five independent measurements was calculated for each set of samples. The tested data were shown in Appendix A. Analysis of variance (ANOVA) was performed on the dataset generated by tensile tests using SPSS software (Appendix A).

The impact strength was conducted on a Tinius Olsen Impact model 503 using the Charpy ISO 179 standard. A “V-notch” was applied with a depth of 2 mm. Seven samples were tested to obtain an average value. The Vicat softening temperature (VST) of the non-annealed and annealed samples was determined using a Vicat softening temperature tester (CEAST VIC/01, Pianezza, Italy) per ISO306:1994. Methyl silicone was used in an oil bath at a heating rate of 120 °C h^−1^ and a specific load of 10 N.

## 3. Results and Discussion

### 3.1. The Relevance of Annealing with PLA

#### 3.1.1. Structure and Morphology of PLA Samples

As shown in Figure 1a and Appendix A, DSC tests on printed non-annealed PLA bars highlighted a *T*_g_ of 58 °C and a large cold crystallinity peak (*T*_cc_) at 98 °C. As PLA has no crystallization peak at a cooling rate of 10 °C min^−1^ due to the poor crystallinity ability, the fast cooling speed during printing would induce amorphous phase formation [8]. Further inspection of Figure 1a showed that the Δ*H*_m_ values were similar for the annealed PLA samples, indicating similar overall crystallinity for the annealed PLA bars. Extending the annealing time or altering the treatment temperature sequentially did not automatically produce a higher *X*_c_, which reached a maximum (about 55%) within 0.5 h annealing at 80 °C or 100 °C. Notably, a small exotherm peak also appeared before the main melt peak. This recrystallization peak is associated with the restructuring of certain existing crystalline structures at high temperatures that involves a solid-state transition from a disordered α’ phase to an ordered α phase, in which the chain packing of the crystal lattice becomes more compacted [36,37]. α phase is the most common and stable crystal structure of PLA.

The non-annealed PLA bars were transparent and became opaque (white) after annealing (Appendix A). Polarizing microscope (POM) images (Figure 1a (inserts)) provide a visualization of the sample structure and morphology and verify the deposition of the material layers, the connection between the layers, and the existence of interior voids. The non-annealed PLA showed very small voids, with an unclear strandline. After annealing, the strands were visible, and small white grains (overlapping fragment crystals) covered the screen. The image for PLA—80 °C 0.5 h was representative of the POM images obtained for samples annealed under other conditions (Figure 1(a2)). Normally, the interface is considered to be a mechanically weaker point in AM polymer materials due to incomplete molecular diffusion. However, the crystal bands, along with the interface (Figure 1(a2)), might be beneficial to the interfacial strength of the polymer [25]. More diffusion of molecules between the layers/strands, followed by crystallization of this small number of the PLA molecules, has been indicated to increase the impact strength [32,38].

XRD measurements can also be used to confirm crystallization behaviors. The non-annealed PLA showed a broad dispersing diffraction peak around 17° (Figure 1b), indicating an amorphous phase with lower crystallinity, which was consistent with the DSC result (*X*_c_ ≈ 20% for non-annealed PLA). The annealed PLA samples were characterized by strong reflection peaks of (200/110) and (203) at 17.3° and 19.6°, respectively, that are consistent with the increases seen with DSC (*X*_c_ > 50% for annealed samples). The peak values were also consistent with values reported in other studies [25,39,40] and were indicative of a typical orthorhombic crystal. Weak reflections at 15.8° and 25.3° belonging to the (010) and (116) plane were also observed.

The smooth and flat cryo-fracture surface of non-annealed PLA highlighted a typical brittle breakage (Figure 1c). In contrast, the surface of annealed PLA is rough and exhibits some fibrillar structures due to the deformation of spherulites, and suggests that more energy is dissipated during the impact fracture [27]. PLA samples, annealed at 100 °C, had a rougher surface relative to those annealed at 80 °C, which is indicative of higher impact strength for the material annealed at the higher temperature. Of course, this still needs to be confirmed, as covered in the next subsection.

#### 3.1.2. Mechanical Properties and VST of PLA Samples

Results for the tensile strength of annealed PLA are shown in Figure 2 and Appendix A together with data for non-annealed samples for comparison. Although the mean value of modulus and tensile stress (maximum load/yield) of PLA increased after annealing, the standard deviation from ANOVA analysis indicated that this increase was insignificant (Appendix A). Meanwhile, the tensile strain of PLA remained low. Consistent with this observation, results of a three-point bending test on EAM PLA annealed at 65–95 °C performed by Wach et al. [29] highlighted that the enhanced crystallinity was not associated with a significant change in the flexural strain.

The notched impact strength of the PLA bars increased significantly after annealing (Figure 2, grey bars; Appendix A). Annealing at higher temperature or prolonging the annealing time further improved the impact strength (Figure 2d, left to right). Closer inspection of the results showed that the notched impact strength of PLA—80 °C 0.5 h was about double the value of neat PLA (Figure 2d). This improvement could be at least partially ascribed to the increase in crystallinity but also to the further perfection of crystallites and interfaces. The crystallinity of PLA—80 °C 0.5 h was 57% compared to the starting value of 20% (Appendix A). The impact strength of PLA—80 °C 2 h was three-fold higher than that of PLA, whereas *X*_c_ of the annealed samples retained a maximum value even with a longer annealing time (Appendix A). For an annealing time of 2 h and an annealing temperature of 100 °C, we observed a notched impact strength that was as high as 15 kJ m^−2^ bearing in mind that the *X*_c_ value of PLA—100 °C 0.5 h was triple that of untreated PLA. As PLA crystallized fastest at around 100 °C, a better-defined network comprising crystals and tie molecules likely forms, promoting a larger plastic deformation that can absorb more fracture energy upon impact load [17]. Fragmented, small size crystals (Figure 1a inserts) can significantly improve the impact strength of PLA by reducing the amorphous portion among spherulites, and by having fewer defects within a spherulite [17,18,41]. Similar results were reported by Yang et al. [17], who demonstrated that the impact strength of PLA 4032D samples exhibited a sharp increase from 3 to 6–8 kJ m^−2^ with higher annealing temperatures (60~90 °C) and longer annealing times (4~8 h).

The VST of non-annealed PLA was ca. 60 °C (Figure 2d, blue bars; Appendix A). Notably, a VST of 160 °C was obtained for the annealed samples, which is consistent with previous studies [15,19,36]. This increase could again be attributed to an increase in crystallinity and the formation of physical crosslinking points in the PLA matrix [16]. Annealing enhances the heat diffusion between layers, also improving the thermal resistance. VST values of printed bars before and after annealing were comparable to bulk PLA processed by other methods, such as injection-made or hot-pressed samples [15,16] that have a VST value of about 60 °C and >140 °C for amorphous and high-crystallinity PLA, respectively.

For completeness, the important properties of the products obtained by various treatments, as well as data for the changes in density and dimensions before and after printing, are presented in Appendix A. The percentage difference in density was within 1% of that for the non-annealed sample, and the dimension changes between pre- and post-annealing samples were less than 3%. The small magnitude of these percentages indicates that printing and dimensional changes should have virtually no effect on the mechanical properties of the sample.

### 3.2. Effect of Blending PLA with PHB

#### 3.2.1. Structure and Morphology of PLA/PHB Blends

PHB is a biopolymer that has high crystallinity and can act as a nucleating agent to induce PLA to crystallize into a more ordered crystalline structure [8]. The high MFI of PHB endows PLA with better flowability, making PHB blending a suitable approach to improve the performance of PLA. As highlighted in previous studies [8,33], both PHB and the PLA/PHB blend crystallize completely at a cooling rate of 10 °C min^−1^. The effect of cooling during printing on the final crystallinity of PLA/PHB blends, covered in the present work, is shown in Figure 3a. As expected, PHB showed no *T*_cc_, indicating its good crystallization ability. The PLA/PHB blend still showed a small recrystallization peak, indicating the blend didn’t crystallize completely after printing, but the *T*_g_ and *T*_cc_ of PLA in the PLA/PHB blend both decreased to 52 and 85 °C, relative to that for PLA alone (58 °C and 98 °C; Figure 1a). As shown in Figure 3b, non-annealed PLA/PHB showed reflection peaks at 10.3° (110_PHB_) [42], 14.4° (020_PHB_) [43,44], and a diffuse peak around 17° (110_PHB&PLA_) [45,46], implying that PHB, but not PLA, crystallized sufficiently during the printing process. For the annealed PLA/PHB samples, *T*_g_ and *T*_cc_ could not be tested due to the frozen molecule chains present after annealing. The reflection peaks in Figure 3b for annealed PLA/PHB displayed a strong reflection at 17.5° (110/200_PLA_) and 19.8° (203_PLA_) [45], reflecting higher crystallinity than the non-annealed sample, which is in accordance with the DSC results. Please refer to the Appendix A for a detailed description of the small peaks for the PLA/PHB blend. Such a blend produced an irregular fracture surface arising from its crystalline structure, whereas pure PLA had a fracture surface that is typical of an amorphous polymer (Figure 3c). In SEM images, cavities due to a debonded, dispersed phase could be seen that indicate poor interfacial adhesion between the two phases. All matrices, shown in Figure 3c, presented a uniform appearance regardless of the annealing time and temperature used for the PLA/PHB samples. The cryo-fracture surfaces of blends were not as rough as those for annealed PLA, indicating lower impact strengths based on the interpretation of the results in Figure 1 and Figure 2.

#### 3.2.2. Mechanical Properties and VST of PLA/PHB Blends

Both the modulus and tensile strength of the PLA/PHB blend declined slightly compared to pure PLA (Figure 4a–c, Appendix A), which could likely be attributed to the lower modulus of PHB and the poor miscibility between PLA and PHB [45]. The PLA/PHB blends had a higher strain at break than PLA due to the introduction of the second phase-PHB crystalline domains that serve as stress concentrators [31,47]. The impact strength of PLA/PHB was somewhat higher than pure PLA (Figure 4d, grey bars; red square). By comparing the SEM image of PLA/PHB (Figure 3c) with that for pure PLA (Figure 1c), more frequent cavitation could be observed as PHB inclusions initiated numerous crazes in the PLA matrix. Crazes should at first significantly improve the impact toughness of PLA, but since PHB is nearly immiscible with PLA, the presence of PHB in the blend should improve the toughness of PLA only slightly. In a study by Bartczak et al. [30], blending of PLA with atactic poly([R,S]-3-hydroxy butyrate) (a-PHB) led to a significant improvement in the impact strength since, unlike PHB, a-PHB is partially miscible with PLA.

The VST of PLA/PHB increased above 150 °C (blue bars) compared to the increase seen at only 60 °C for non-annealed PLA (Figure 4d, red circle). The PLA/PHB blend showed a smaller Δ*H*_cc_ than PLA, which indicates a higher *X*_c_, but according to the XRD data, the non-annealed PLA/PHB sample did not sufficiently crystallize (i.e., a diffuse peak near to 17°) during printing. Thus, the high VST could be ascribed to the favorable crystallization property of PHB at the slow heating rate of 120 °C min^−1^ used with the VST tester. PHB thus acted as a nucleating agent in the PLA matrix and promoted rapid crystallization during testing and, in turn, resulted in an increased VST.

Usually the ductile behavior of PHB material would seriously deteriorate with progressive crystallization; however, by using a simple annealing treatment, PHB can be rejuvenated while subsequent aging is prevented to a large extent [48,49]. In this study, PLA/PHB was annealed in consistence with the annealing of PLA to see the effect of annealing. The tensile modulus increased significantly after annealing, as the increased crystallinity strengthened the material. This result is similar to that of Takayama et al. [14], who found that the bending modulus and strength of PLA/polycaprolactone (PLA/PCL) were effectively improved by annealing since crystallization of the PLA phase strengthened the structure of the PLA/PCL blend. In contrast, the tensile strain of the PLA/PHB blend is decreased relative to PLA alone because the tighter structure of the crystals is not amenable to ductile deformation [14]. A reduction in the compatibility caused by annealing could also diminish the tensile strain of the product. However, unlike for PLA, the impact strength of the PLA/PHB blend showed an unexpected minor increase after annealing that might be due to the restricted deformation of PLA and PHB arising from a large number of crystallites. In contrast to homogenous PLA, poor interfacial adhesion in the immiscible PLA/PHB blend limited the amount of improvement in impact strength. As the VST value was also rather high for the PLA/PHB blend (>150 °C), the annealing treatment appeared not to be necessary to further tune this property.

Taken together, these results indicated that blending with PHB did improve the impact strength, tensile strain, and VST of PLA, but annealing only slightly influenced the mechanical properties of the blend. As such, annealing is not necessary to improve these properties, but future studies may confirm the usefulness of annealing to blend with a compatibilizer [50].

## 4. Conclusions

To improve the thermal and mechanical properties of PLA produced by extrusion-based additive manufacturing, we used several different experimental techniques to examine the effects of annealing and blending with PHB both separately and together.

Results for DSC and XRD showed the difference between non-annealed and annealed PLA, that annealing significantly increased the crystallinity of PLA. POM images of non-annealed and annealed PLA showed a significant difference manifested as clearer strands for annealed PLA arising from the enhanced crystallization following this treatment. High crystallinity and the crystals along the interface contributed to the higher impact strength. SEM images showed that non-annealed PLA had a smooth surface that became rough after annealing. This result was consistent with the mechanical testing results, as the notched impact strength of a PLA bar increased significantly after annealing. Annealing at a higher temperature or prolonging the annealing time could improve the impact strength even further. Vicat softening temperatures of printed bars before and after annealing were also comparable to bulk PLA processed by other methods, such as injection-molding or hot-pressing.

DSC and XRD results showed that the presence of PHB increased the crystallization rate and crystallinity of the PLA/PHB blend. However, a maximum crystallinity was reached after a certain annealing time. The fracture surface of PLA/PHB was rougher than that of neat PLA, and it remained coarse after annealing, in accordance with the minor increase seen for impact strength. Blending PHB with PLA allowed the printing temperature to be lowered and also improved the product toughness and Vicat softening temperature. PHB was thus a good supplement to enhance the flowability and thermal properties of PLA. The PLA/PHB blend showed a lower tensile modulus and stress compared to PLA alone due to the intrinsic lower stiffness of PHB and the rather poor miscibility between PLA and PHB. Meanwhile, the strain at break increased upon the introduction of a second phase. The modulus and strength of the PLA/PHB blend improved considerably after annealing, due to the strengthened structure of the PLA phase promoted by annealing-mediated crystallization. The presence of PHB did improve PLA’s VST, which is maintained at a high level for the PLA/PHB blend before and after annealing through the facilitation of crystallization in the blend by PHB.

To summarize, the impact strength and VST of PLA could be significantly improved by annealing, and PHB endowed PLA a high VST and better toughness without post-treatment, thus saving time and energy.

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
