# Peer review of "Improving Mechanical Properties for Extrusion-Based Additive Manufacturing of Poly(Lactic Acid) by Annealing and Blending with Poly(3-Hydroxybutyrate)"

_polymers, 2019, doi:10.3390/polym11091529_

Round 1

Reviewer 1 Report

The manuscript is well written and the data are adequately commented. A large number of characterisations are used to investigate the different produced formulations. 

I suggest the publication in Polymers, but some clarifications are necessary to improve the quality of the article. 

Minor revision. 

specifically: 

Introduction- section: The authors are invited to explain the novelty and the final  application sector of this research DSC analysis: The authors are invited to comment better the effect of PHB in the different formulations and the annealing process ( figure 1 and 3). 

Reviewer 2 Report

Manuscript ID: polymers-588359
Title: Improving mechanical properties for extrusion-based additive manufacturing of poly(lactic acid) by annealing and blending with poly(hydroybutyrate)
Authors: Sisi Wang , Lode Daelemans , Rudinei Fiorio , Maling Gou , Dagmar R. D’hooge , Karen De Clerck * , Ludwig Cardon *

Authors described methodology, how thermal properties of PLA based thermoplastic materials can be improved for 3D printing applications.

The article is well written with clear methodology and explanations, methods are described satisfactorily.

Comments to the article:

I have only one note to the presented results. It is well known rather slow physical ageing of PHB materials due to post crystallization. Thus, external properties of materials can change over time. This feature was not considered or was not discussed in the introduction eventually in discussion part of the article. Could you please comment this issue?

In conclusion, I do recommend to accept this article for Polymers journal after revision according to comments.

Author Response

Answer to reviewer 2:

Usually, the biodegradable material PLA and PHB are not aimed for durable use, explaining why the aging phenomenon is not focused on here. Although the ductile behavior of PHB material would seriously deteriorate with progressive crystallization, by using a simple annealing treatment PHB can be rejuvenated while subsequent aging is prevented to a large extent [Koning, G. J. M. d., 1993]. In this study, PLA/PHB was annealed in consistence with the annealing of PLA, finding impact strength of blends increased slightly after annealing. This indication was added in PLA/PHB’s mechanical performance section (line 304 and following).

Reference: Koning, G. J. M. d.; Lemstra, P. J. Crystallization phenomena in bacterial poly[ ( R)-3- hyd roxybutyrate]: 2. Embrittlement and rejuvenation. polymer 1993, 34, 19, 4089-4094.

Reviewer 3 Report

Dear authors,

This work brought to us interesting results about the effect of annealing and blending with PHB on PLA polymers applied to 3D printers. The work present high quality, however, I suggest some modifications before the acceptance.

Abstract: I think should be improved. It is not clear the objective of the study and the conclusions are only comments about the results. In the same way, it is difficult to figure out what the meaning of your results, how they contributed to your objective. Mechanical tests, POM and VICAT were also used to characterize the materials and processing.

It would be great if the authors include some pictures from the printed specimens to see the quality of the printed components, and if they change the aspect after the annealing.

Lines 157 to 160 – It would be great a better explanation about alfa-phase.

Lines 175 to 177 - What kind of molecules you say here?

The authors talk about mechanical behaviour in the part where they present DSC, X-Ray results. Maybe, you can consider show us the mechanical properties before.

Why the annealing time nor temperature does not affect the surface morphologies in the blends?

Conclusions: Regarding lines 322 and 323, I think only printed samples were showed in this work. What would be a general conclusion for yourwork? Annealing or blending? What wasthe best for printed components?

Minor:

Please, check in the entire manuscript:

Please, correct PHB in the title Keep the same style for “poly(lactic acid)” Please use the notation wt.% instead m% POM isn’t described before scheme 1 Please, check all references citation in the manuscript: use “… polymer [25].” instead “…polymer.[25] “ Please, check Xc.
